# Urban-rural inequalities and spatial arrangement of informed choice of family planning in Ethiopia: Further analysis of 2016 Ethiopian demographic health survey

**Abiyu Abadi Tareke**[1]\*, **Bayley Adane Takele**[2], **Mohammedjud Hassen Ahmed**[3], **Masresha Derese Tegegne**[4], **Habitu Birhan Eshetu**[5]

**1** Amref health Africa in Ethiopia, SLL project, Covid-19 vaccine /EPI Technical Assistant at West Gondar Zone, Gondar, Ethiopia, **2** Department of Epidemiology and Biostatistics, Institute of Public Health, College of Medicine and Health Sciences, University of Gondar, Gondar, Ethiopia, **3** Department of Health Informatics, Mettu University, Metu, Ethiopia, **4** Department of Health Informatics, Institute of Public Health, College of Medicine and Health Sciences, University of Gondar, Gondar, Ethiopia, **5** Department of Health Education and Behavioral Sciences, Institute of Public Health, College of Medicine and Health Sciences, University of Gondar, Gondar, Ethiopia

\* abiyu20010@gmail.com

**Data Availability Statement:** All relevant data are available from: https://www.dhsprogram.com/.

## Abstract

### Background

Ethiopia has made satisfactory progress in improving maternal and child health over the past two decades. The introduction of family planning through informed choice is one of the main strategies to improve maternal and child health. However, this positive progress may have masked the significant urban-rural disparities in informed choice for family planning.

### Objective

To identify factor contributing to observed urban-rural disparities and to determine the spatial distribution of informed family planning choices in Ethiopia.

### Methods

The study used information from 3,511 women currently using contraceptives (rural-2685 and urban-826) as per recent Ethiopian demographic health survey cross-sectional data. Spatial and descriptive, bivariable, and multivariable logit-based decomposition analysis methods were used.

### Results

The spatial configuration of uninformed choice was clustered. The primary cluster (LLR = 34.8, p-value<0.001) was located at the southern portion of Amhara region that covers east & west Gojjam, south Gondar and south Wollo administrative zones. The magnitude of informed choice was 12 percent higher in urban residents compared to rural residents. Urban-rural gap was attributed to variations in characteristics (74%). Place of family

**Funding:** The author(s) received no specific funding for this work.

**Competing interests:** The authors have declared that no competing interests exist.

**Abbreviations:** EDHS, Ethiopian Demographic health survey; LLR, Log-Likelihood Ratio; FP, Family Planning; NGO, Non-Governmental Organization; GPS, Global Positioning System; RR, Relative Risk; IQR, Inter-Quartile Range; EA, *enumeration areas*; SNNP, Southern Nations, Nationalities, and People and; SMS, Short Message Service.

planning offer i.e., private health facility, being aged between 35 and 49 years, and having visited to health facility in the last 1 year are found decrease the urban-rural gap of informed family planning choice by 15%, 9% and 5% respectively. Conversely, being aged between 25 and 34 years, being a listener to radio has increased the gap by 9% and 12% respectively.

## Conclusion

The variables being private health facility visitors, being aged between 35 and 49 years and having visited health facilities in the last one year are found to increase the gap of informed family planning choices between urban and rural residents Besides, the spatial distribution of uninformed family planning choices is non-random.

## Introduction

Family planning is the a technique for either limiting number of children, [1] or want to delay their next birth [Spacing] [2, 3] after having unprotected sexual intercourse. Informed choice of family planning is defined as when a woman chooses a family planning method after receiving information about the possible side effects, what to do in case of side effects happen, and possible alternative methods by healthcare providers [1, 4]. If there are no contraindications, the choice of family planning (FP) methods is ultimately at the decision of the user [5].

Clients are entitled to make voluntary, informed decisions about family planning services based on options, knowledge, and comprehension [6]. Ethiopian national family planning guideline appreciate health care professionals to provide informed decision making as the best strategy to quality family planning services [7]. In sub-Saharan Africa countries, the level of informed choice of family planning is not satisfactory [6]. Potential African countries are providing family planning services without informing possible side effects. The level of informed choice of family planning is very low in Burundi [35%], Niger [38%], Gambia [42%], Benin [47%] and Mali (48%) [6]. In Ethiopia, the level of informed decision of family planning is poor (25%) compared to other African countries [6]. Recent research has confirmed that the extent of informed choice for family planning methods is inconsistent with the place of residency in Ethiopia [6]. Women residing in urban areas were 1.4 times more likely to be informed about family planning methods than women living in rural slums. However, the factors responsible for the disparity of uninformed choice between the two groups (urban and rural) are not examined. Education about the side effects of family planning and other alternative methods plays a significant role in the uptake of long-acting methods [8–11]. The low use of long-acting family planning methods may be related to uninformed family planning choices.

To our knowledge, and after a thorough review of literatures, previous studies in Ethiopia have investigated the magnitude and socio-demographic factors associated with uninformed family planning choices [12, 13]. However, none of these studies examined the extent of urban-rural inequalities. Moreover, none of these studies identified the factors contributing to the urban-rural gap in the informed family planning decision. In addition to that, this study determined the spatial distribution of uninformed choices of family planning in Ethiopia and identified areas exhibiting elevated aggregation of uninformed choices (popularly known as hotspots], which facilitate the formulation of interventions tailored to specific locations.

Therefore, the objective of this study is to identify and quantify the factors that contribute to the observed urban-rural gap in informed family planning decisions. Besides, information obtained from Georeferenced datasets could help program developers and local decision-makers develop location-specific strategies to address the problem of uninformed family planning choices, rather than indiscriminately providing services to the entire part of the country.

## Methods and materials

### Study design and period

This study was conducted using information from the fourth cross-sectional demographic health survey of Ethiopian (EDHS) communities, which was carried out between January 18 and June 27, 2016.

### Study area

Ethiopia is the second-most populous country in Africa. The country's overall population was expected to reach roughly 103,000,000 by 2021, according to the Population and Housing Census (PHC) projection from 2007. According to the 2007 Census, the majority of people lived in rural regions (83.6%), there were 4.7 people per family on average, and 47% of all women were between the ages of 15 and 49 [14].

### Population

**Source population.**   All reproductive age women in Ethiopia.

**Study population.**   Women aged 15–49 years who were modern family planning users during the data collection period.

**Dependent variable.**   The outcome variable for this study is informed choice of family planning for the decomposition analysis and uninformed family planning choice for the spatial analysis.

**Independent variables.**   According to our extensive literature reviews, maternal age, maternal education (none, primary, secondary, higher), marital status, occupation, wealth status (poorest, poor, middle, rich, and richest), sources of contraception (private, non-governmental organizations (NGO), pharmacy, other), marital status (never married/ever married), family planning message on mobile phone (got messages/ not receive messages), Working status (working/ no working), visit a health facility in the last 12 months, frequency of reading newspapers or magazines (not at all, less than once a week, at least once a week), frequency of listening to the radio (not at all, less than once a week, at least once a week), and Frequency of watching television (Not at all, Less than once a week, At least once a week) were considered as independent variables for this study.

**Operational definitions.**   *Informed choice of family planning.* women of reproductive age who used contraceptive methods but were not informed of side effects, what to do if they experienced the side effects or problems, and who were not informed of other contraceptive methods that could be used, were labeled as "uninformed choice" with code 1, otherwise as" informed choice" with code 0.

*Media exposure.* created by combining whether a respondent reads the newspaper, listens to the radio, and watches television. If the respondent was exposed to at least one of the three media this is labeled "exposed" and coded" 1", otherwise "not exposed" is coded "0".

**Data source.**   We used the Kids' record (KR) dataset of the 2016 EDHS to further analyze.

**Data collection tools and procedures.**   The website of the DHS measure (http://www. dhsprogram.com) was employed to effectuate registration for access to the 2016 Ethiopian

DHS Datasets and Global Positioning System (GPS) data, with requisite permission having been duly procured in order to gain access to the requested tools. Accordingly, all required data were downloaded from the Demographic and Health Surveys [15] Program website. A global positioning system was used to collect the geographic coordinates of each cluster [15].

**Data quality control.**   The data collectors of this survey documented that questionnaires were pre-tested in all three local languages (Amharic, Afaan Oromo, and Tigrigna) to ensure that questions were clear and understandable to respondents [1].

**Sample size and sampling procedures.**   A two-stage, stratified, clustered sampling procedure was implemented across all 11 (now expanded to 13) geographic administrative areas comprising 9 regions and 2 city administrations. The first strata included a total of 645 enumeration areas (EAs) that were selected proportionally to the EAs size of the nine geographical regions and two administrative cities. In the second strata, every eleven administrative divisions were further sub-grouped into urban and rural residents, yielding a total of 21 sampling strata. From each cluster, 28 households were chosen using an equal probability technique. This study ultimately includes a weighted sample of 3511 participants [1, 16, 17].

**Data management.**   Data were processed, reviewed, sorted, and recoded using STATA/SE version 16.0. To account for the effects of the survey's complex sampling design or the hierarchical nature of the EDHS dataset, to restore survey representativeness, and to obtain reliable statistical estimates, the data were weighted via applying the STATA command "svyset." This command was prepended to each analysis in this study.

Arc GIS version 10.8 software was used to visualize the spatial distribution and locate hotspot areas (clusters).To measure the deviation of the spatial arrangement of the uninformed choice from randomness, the global spatial autocorrelation (Global Moran's I) was calculated [18, 19]. A positive value of Moran's I represent positive spatial autocorrelation (cluster together), whereas a negative value indicates dispersed arrangement.

**Logit-based multivariate decomposition analysis.**   To find factors that contributed to the gap of informed choice of FP between urban and rural residents, multivariate decomposition analysis was calibrated. This analysis utilizes the output from the logistic regression model to assign the observed change in informed choice of FP rate between urban and rural through subdividing into components (i.e., due to endowment and coefficients).

For logistic regression, the Logit or log-odd of difference between urban and rural is taken as:

Logit (rural)-Logit (urban) = F (X urban*rural)–(F Xurban*βrural)

= [(F(X urban*β2rural)-F (Xurban*βrural)/E)] + [(F(X urban*βrural)-F (Xurban*βrural)/C)] [20].

X indicates the outcome variable i.e., informed choice of FPBeta (β) indicates that, regression coefficient of each selected independent variables

The E component refers to change in informed choice of FP imputable to differences in endowments or characteristics. The C component refers to a change in informed choice of FP imputable to differences in coefficients or effects. During decomposition analysis the categories urban and rural were recorded as "0" and"1" respectively. Percent contributions with a 95% confidence interval (CI) of coefficients and a p-value <0.05 were reported. Candidate variables for multivariable decomposition regression were selected according to the p-value from bivariable decomposition regression. Those variables with p-value less than 0.2 in the bivariable regression were moved to multivariable decomposition regression analysis.

**Spatial scan statistical analysis.**   Bernoulli model and purely spatial Kulldorff's scan statistical analysis was deployed to detect clusters (areas with concentrated numbers of women making uninformed choices FP). If the spatial pattern of uninformed choice of FP is randomly distributed across space, it is not doubtful that the development of site-specific strategies will be effective in curbing the high prevalence of uninformed choice.

Only areas with a high rate of uninformed choices of FP were applied to determine the geographical location of statistically significant clusters using SaTScanTM v10.0.1 software. We used the Bernoulli model because the data were binary (uninformed or informed choice). Women who were uninformed about the choice of FP were considered cases (1), whereas those who were informed were considered non-cases (0).

The case file (1), non-case file (0), and coordinate files (latitude and longitude) were imported into SaTScan $^{TM}$ software to determine the location of significant clusters. The maximum size of the scan window was scaled according to the percentage of the total population at risk. The maximum radius of the circle was adjusted to be less than 100 km. This is done to facilitate the development of intervention strategies, where clusters with optimal radius are easier to manage than large clusters.

To avoid overlooking very small and very large clusters, the maximum geographic cluster size was adjusted to < 50% of the population at risk as an upper limit. The most likely (primary), cluster was determined using p-value and likelihood ratio tests. The cluster with the highest likelihood ratio represented the most likely cluster [21], and the remaining clusters with statistically significant log-likelihood ratios (LLR) were designed as possible secondary possible clusters. The relative risk (RR) parameter of the uninformed choice of FP in each cluster was calculated to estimate the risk of uninformed choice within the cluster areas [22, 23].

### Ethics approval and consent to participate

As the authors of this manuscript did not involve collection of data from the participants, consent to participate and ethical approval is not required. But the authors have granted to access the dataset of 2016 EDHS through registering to www.dhsprogram.com website. The data granted from this website have no personal identifying information (anonymous) and the participants' confidentiality and privacy issue is not an inconsequential in this scenario. All the methods were performed in accordance with the relevant guidelines and regulations of DHS measures.

### Results

A total of 3511 (weighted) modern family planning users were included in this study. The median age of the study participants was 28 years (IQR = from 23 to 34 years). As shown in the Figure (below), only 40.7% [95% confidence interval: 39.12% to 42.4%] of current contraceptive users were informed about FP choices. Additionally, the magnitude of informed choice was 12 percent higher among urban residents than rural residents (Fig 1).

The greater number of study participants were from rural slums (76.5%). More than 60% of the study participants were from Oromia and Amhara regions. Of all participants, 1838 (52.3%) were orthodox Christian followers, accompanied by Protestants (25%) (Table 1).

### The spatial arrangement of the uninformed choice of FP

The spatial arrangement of the uninformed choice of FP in Ethiopia was not evenly distributed. The null hypothesis of global spatial autocorrelation assumes that the uninformed choice of FP was randomly distributed across the study area. The positive z-score, positive Moran's index, and statistically significant p-value from Fig 2 prove that the spatial arrangement of uninformed choice was clustered. The corresponding p-value of less than 0.01 on the right side of the figure can be interpreted to mean that the probability of randomness of the observed spatial pattern (i.e., clustering) of uninformed choice is less than 1% (Fig 2).

Fig 3 shows hotspot areas (areas with an overwhelming number of women who have uninformed FP choices) among contraceptive users in Ethiopia. the primary cluster (shaded in red)

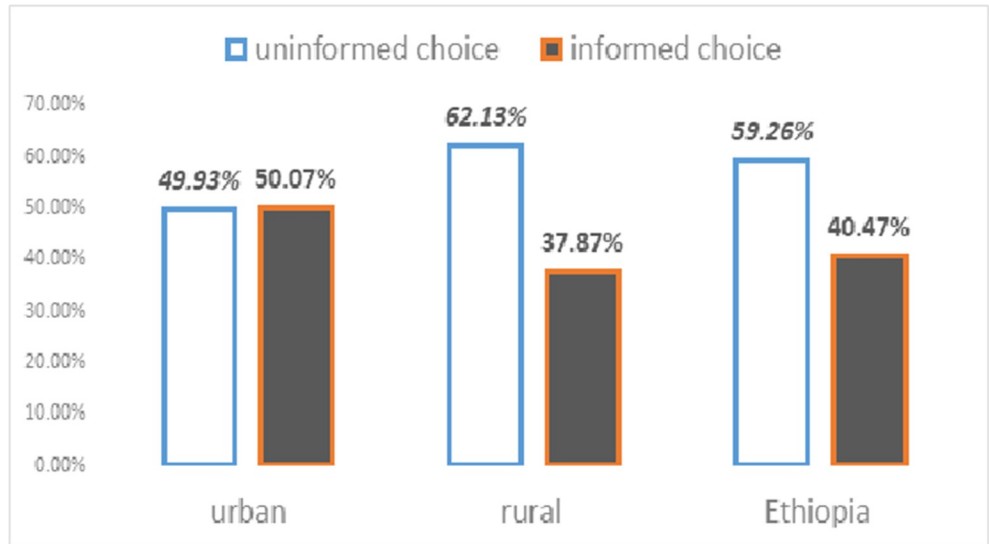

**Fig 1. Distribution of informed family planning choices by place of residence in Ethiopia, 2016.**

(LLR = 34.8, p-value<0.001) focused primarily on the southern part of the Amhara region, which included East & West Gojjam, South Gondar, and south wollo zones and was centred at the latitude of 710.984556˚N, and longitude of 38.044450˚E with a radius of 97 km (Fig 3 and Table 2).

## Geographic clustering of uninformed choice

Fig 3.

## Decomposition analysis

Variables with a p-value of less than 0.2 from the bivariable decomposition analysis were picked as the contender variable for the multivariable decomposition analysis. The composition factors of region and religion were not included in the multivariable composition analysis because of having a p-value greater than 0.2 at bi-variable decomposition analysis. The multivariable decomposition regression models revealed that 74% of the changes in informed choice of FP could be attributed to changes in population characteristics (population dynamics) (Table 3). This can be interpreted to mean that if household characteristics related to the explanatory variables had been equalized between urban and rural residents, the Urban-rural gap of informed choice of FP would have declined by 74%. However, the change due to the coefficients is not statistically significant and the detailed result of the decomposition of each variable is not included in the table.

**Non-linear decomposition of informed choice of FP.** From the multivariable non-linear decomposition analysis Table 3, the age group between 35 and 49 years, receiving the last FP at private health facilities, and being visited health facilities in the last year were the gap tightening variables (negative percentage contribution). Conversely, the age group between 25 and 34 years, lack of access to FP-related mobile short message service (SMS) and listening to the radio were variables that increased the difference in informed choice of FP between these two groups (Table 3).

As can be seen from the table (above), the difference between urban and rural areas is mainly explained by the location of FP supply (i.e., private facilities), followed by listening to

**Table 1. Socio-demographic characteristics of study participants in Ethiopia, 2016.**

| Variables | Unweighted frequency (n = 2,862) | Weighted frequency (n = 3,511) | Percent of weighted frequency |
|---|---|---|---|
| **Choice of family planning** | | | |
| Informed choice | 1270 | 1430 | 59.30 |
| Not informed choice | 1592 | 2081 | 40.70 |
| **Age** | | | |
| 15–24 | 868 | 27 | 27.21 |
| 25–34 | 1,337 | 1676 | 47.74 |
| 35–49 | 657 | 879 | 25.05 |
| **Place of residence** | | | |
| Urban | 1095 | 826 | 23.52 |
| Rural | 1777 | 2685 | 76.48 |
| **Religion** | | | |
| Orthodox | 1573 | 1838 | 52.34 |
| Protestant | 589 | 824 | 25.48 |
| Muslim | 665 | 725 | 20.64 |
| Other | 35 | 54 | 1.54 |
| **Educational status** | | | |
| None | 1,172 | 225 | 6.41 |
| Primary | 1,052 | 1804 | 51.39 |
| Secondary | 377 | 1148 | 32.71 |
| Higher | 261 | 333 | 9.49 |
| **Work status** | | | |
| Working | 1,631 | 1,919 | 54.65 |
| No working | 1,231 | 1,592 | 45.35 |
| **Region of residency** | | | |
| Tigray | 375 | 265 | 7.54 |
| Afar | 73 | 12 | 0.33 |
| Amhara | 532 | 1137 | 32.39 |
| Oromia | 353 | 1057 | 30.1 |
| Somali | 14 | 4 | 0.11 |
| Benishangul–Gumuz | 227 | 32 | 0.92 |
| SNNP | 447 | 793 | 22.6 |
| Gambela | 188 | 10 | 0.28 |
| Harari | 150 | 6 | 0.18 |
| Addis Ababa | 330 | 180 | 5.13 |
| Dire Dawa | 173 | 265 | 7.54 |
| **Wealth status** | | | |
| Poorest | 313 | 396 | 11.28 |
| Poorer | 406 | 638 | 18.16 |
| Middle | 456 | 730 | 20.79 |
| Richer | 478 | 775 | 22.09 |
| Richest | 1,209 | 972 | 27.68 |
| **Marital status** | | | |
| Never married | 88 | 66 | 1.87 |
| Ever married | 2,774 | 3445 | 98.13 |
| **family planning message on mobile phone** | | | |
| Got message | 73 | 81 | 2.30 |
| Not receive message | 2,789 | 3430 | 97.70 |

*(Continued)*

**Table 1.** (Continued)

| Variables | Unweighted frequency (n = 2,862) | Weighted frequency (n = 3,511) | Percent of weighted frequency |
|---|---|---|---|
| **place of FP offer** | | | |
| Private | 2,654 | 3,363 | 95.8 |
| NGO | 79 | 43 | 1.22 |
| Pharmacy | 115 | 92 | 2.61 |
| Other | 14 | 13 | 0.37 |
| **Visit to health facility in the last 12 months** | | | |
| Not visited | 1,893 | 2,335 | 66.52 |
| Visited | 969 | 1,176 | 33.48 |
| **Frequency of listening to radio** | | | |
| Not at all | 1,738 | 2,309 | 65.78 |
| Less than once a week | 543 | 575 | 16.39 |
| At least once a week | 581 | 626 | 17.83 |
| **Frequency of reading newspapers or magazines** | | | |
| Not at all | 2,378 | 3,115 | 88.71 |
| Less than once a week | 368 | 297 | 8.46 |
| At least once a week | 116 | 99 | 2.83 |
| **Frequency of watching television** | | | |
| Not at all | 1,670 | 2,502 | 71.27 |
| Less than once a week | 397 | 460 | 13.09 |
| At least once a week | 795 | 549 | 15.64 |

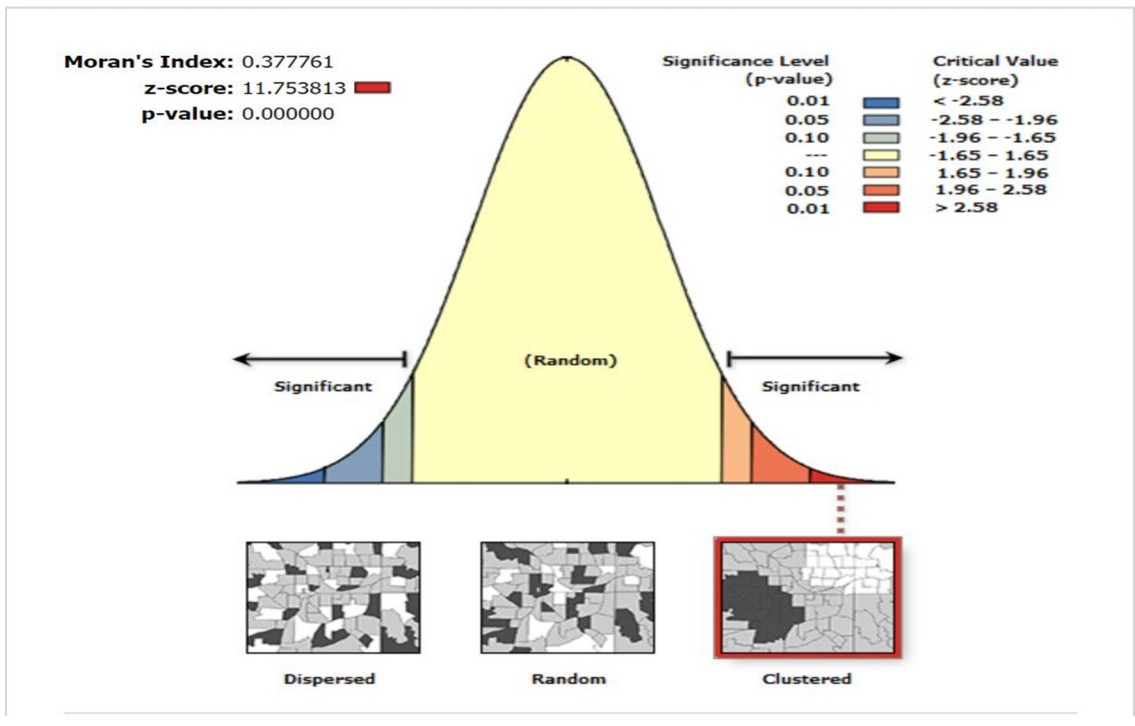

**Fig 2. The spatial arrangement of uninformed choice of FP in Ethiopia, 2016.**

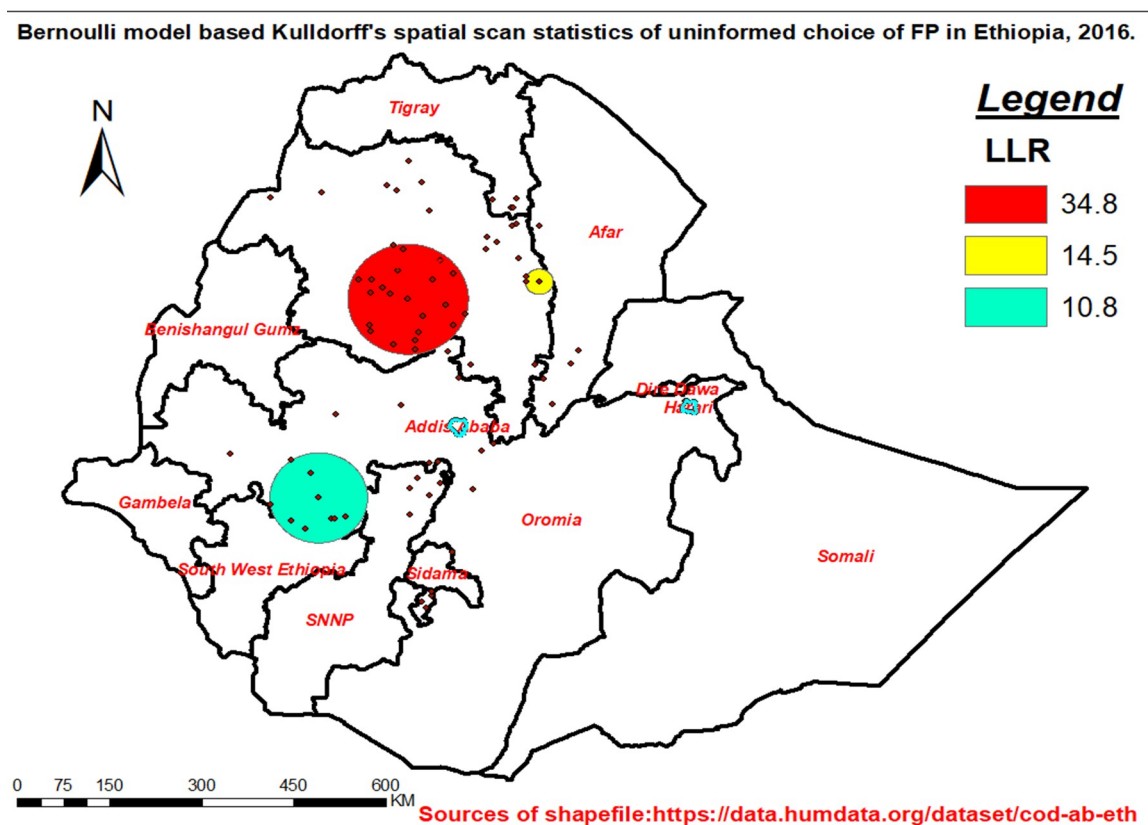

**Fig 3. Spatial scan of boundaries with the highest number of uninformed choices of FP in Ethiopia, 2016.**

radio programs. In addition, the age of current users of FP also had a strong influence on the urban-rural gap in informed choice. To illustrate, if the composition (characteristics) of where FP is offered (in private facilities) were identical in urban and rural areas, the difference in informed choice of FP in Ethiopia would have increased by 16 percent.

Similarly, if the composition (characteristics) of women aged 35–49 had remained the same for urban and rural participants, the difference in the extent of informed choice would have

**Table 2. A spatial cluster of uninformed choice of family planning in Ethiopia, 2016.**

| Outputs | Primary cluster | Secondary cluster 1 | Secondary cluster2 |
|---|---|---|---|
| Expected cases | 222 | 24 | 77 |
| Observed[a] | 295 | 39 | 102 |
| EA[b] | 22 | 3 | 9 |
| latitude | 10.984556 N | 11.260606 N | 7.858150 N, |
| longitude | 38.044450 E | 39.965066 E | 36.733552 E |
| Cluster RR[c] | 1.38 | 1.62 | 1.34 |
| Window Radius | 97 km | 22.31 km | 78.9 km |
| Cluster LLR[d] | 34.8 | 14.5 | 10.8 |
| p-value | <0.001 | <0.001 | <0.01 |

*Note: a = See Fig 3 for spatial pattern of spatially large clusters, b = number of enumeration areas (EA) incorporated, c = relative risk d = log-likelihood ratio of each cluster.*

**Table 3. Decomposition of informed choice of FP urban-rural differences in women using 2016 EDHS.**

| Informed choice | Coefficient | p-value | 95% CI of coefficient | | Percent contribution |
|---|---|---|---|---|---|
| **Characteristics (E)** | **-0.105** | **0.016** | **-0.190** | **-0.020** | **74%*** |
| Coefficient (C) | -0.036 | 0.436 | -o.130 | 0.056 | 26% |
| Total Difference (R) | -0.141 | 0.000 | -0.178 | -0.106 | |
| Differences in Characteristics (E) | | | | | |
| **Age group** | | | | | |
| Based group (15–24) | | | | | |
| **25–34** | **-0.01305** | **0.000** | **-0.020** | **-0.006** | **9.2007**** |
| **35–49** | **0.013172** | **0.002** | **0.005** | **0.022** | **-9.2886*** |
| **Wealth quintile** | | | | | |
| Based group (Poorest) | | | | | |
| Poorer | -0.004 | 0.655 | -0.02 | 0.012 | 2.5899 |
| Middle | 0.0022 | 0.811 | -0.016 | 0.02 | -1.5271 |
| Rich | 0.0086 | 0.312 | -0.008 | 0.025 | -6.0639 |
| Richer | -0.057 | 0.172 | -0.140 | 0.025 | 40.038 |
| **Religion** | | | | | |
| Based group (Orthodox) | | | | | |
| Protestant | 0.0060 | 0.209 | -0.0033 | 0.015 | 4.1588 |
| Muslim | -0.000055 | 0.196 | -0.0001 | 0.000 | 0.038617 |
| Other | -0.000052 | 0.880 | -0.0004 | 0.0006 | 0.036803 |
| **Educational status** | | | | | |
| Based group (Higher) | | | | | |
| None | -0.02631 | 0.52 | -0.1064 | 0.0538 | 18.551 |
| Primary | 0.004616 | 0.401 | -0.006 | 0.0154 | -3.2551 |
| Secondary | 0.001435 | 0.936 | -0.0334 | 0.036 | -1.012 |
| **Working status** | | | | | |
| Based group (Not working) | | | | | |
| Working | 0.00028 | 0.852 | -0.0027 | 0.0032 | -0.19789 |
| Marital status | | | | | |
| Based group (Never married) | | | | | |
| Ever married | -0.00065 | 0.879 | -0.0090 | 0.0077 | 0.45998 |
| **Access to FP related mobile SMS** | | | | | |
| Based group (Accessed) | | | | | |
| No access | **-0.00875** | **0.045** | **-0.0173** | **-0.0002** | **6.1671*** |
| **Visit to health facility in the last 12 months** | | | | | |
| Based group (Not visited) | | | | | |
| Visited | **0.00719** | **0.0000** | **0.0049** | **0.0095** | **-5.067**** |
| **Place of FP offer** | | | | | |
| Based group (Government) | | | | | |
| Private | **0.02257** | **0.023** | **0.0031** | **0.0420** | **-15.916*** |
| **frequency of reading newspaper or magazine** | | | | | |
| Based group (Not at all) | | | | | |
| Exposed | -0.02426 | 0.103 | -0.0535 | 0.005 | 17.108 |
| **frequency of listening to radio** | | | | | |
| Based group (Not at all) | | | | | |
| Exposed | **-0.01667** | **0.036** | **-0.0322** | **-0.0011** | **11.756*** |
| **frequency of watching television** | | | | | |
| Based group (Not at all) | | | | | |

*(Continued)*

**Table 3.** (Continued)

| Informed choice | Coefficient | p-value | 95% CI of coefficient | | Percent contribution |
|---|---|---|---|---|---|
| Exposed | -0.0166 | 0.617 | -0.0817 | 0.0485 | 11.704 |

**Annotation:** the boldface texts indicate statistical significance, the single asterisk indicates statistical significance at p-value <0.05, the double asterisk p-value <0.01, and the triple asterisk p-value< 0.001.

increased by 9.3%. Oppositely, the gap in informed choice of FP between rural and urban residents would have been narrowed by 9%, if the composition of women aged between 25 and 34 years was the same between rural and urban in 2016. Besides, the gap would have been narrowed by 11 percent, if the composition of radio listeners stayed the same between the two groups.

## Discussion

This study found that the spatial arrangement of uninformed choice of FP is clustered. Moreover, variables like woman's age, place of FP offer, media exposure, access to mobile SMS, and visit to health facilities are among the contributing compositional variables to the gap of informed FP choice between urban and rural residents. The current study concentrated on the quantitative analysis of the informed choice of FP data from a spatial vantage point using cluster analysis. Using the local spatial statistics (SaTScan), the spatial arrangement of uninformed choice was concentrated in east & west Gojjam, south Gondar and wollo zones of the Amhara region. These provinces were also identified with the poor practice of health communication on family planning [24]. Moreover, earlier research also documented that informed choice of FP varied across regions and places of residency [25]. A possible explanation for that inequality might be, women residing in urban areas are highly likely to get in touch with well trained and experienced health professionals who can nourish theme information related to the choice of FP. Tailoring plans and programs based on geographic locations would be more effective and resource-saving instead of random health service provisions to all areas of the country.

Moreover, women living in rural provinces were laggards compared to their urban counterparts in having informed choices of FP. Those results are consistent with another similar prior study in the context of India [26]. These results are likely to be related to rural women who might have less opportunity to access well trained health professionals who give adequate information about methods of choice. Rural health facilities in Ethiopia has less knowledgeable and less autonomous in their choice of health care services [27]. It would be interesting to focus on rural residing women in offering information related to the choice of family planning.

Interestingly, getting family planning services at private health facilities is found to narrow the gap in informed choice between rural and urban residents. This finding was in agreement with other studies [12, 28]. The reason for this could be the client flow at private health facilities are lower than at public health facilities so that informed choice of family planning can be practically achieved in private health facilities. Moreover, since private health facilities are commercial, they could have a good client handling approach including delivery of detailed information about the service and treatment they are providing to their clients [29–31]. It is better to expand private health facilities to rural areas as equal as the public health facilities to have informed choice services of FP. Compared to the age group of 25 to 34 years being age group between 35 and 49 years is found decrease the gap in informed choice of family planning between rural and urban residents. Other studies also reported similar findings [30, 32–34]. The possible reason behind this could be elder mothers from rural areas may also seek

more information about the family planning method that could be delivered to them because they might experience side effects and discomforts in their prior utilization of family planning methods.

Unsurprisingly, visit to health facilities in the last 12 months were found to narrow the urban to rural gap in the informed choice of family planning, and it was consistent with another study done in Ethiopia [12]. This could be due to the fact that rural resident women might remember their informed choice of family planning if they visit health facilities recently but rural resident women who had not visited health facilities recently may forget about whether they were offered informed choice of family planning services. On the hand, urban women are more likely to remember the informed choice of family planning services as they are more educated and exposed to media than their rural counterparts. Additionally, information flow and access are increasing over time which could make family planning providers in rural health facilities to be more aware of the right of being informed during the choice of their family planning method [35].

This study showed that Access to FP related mobile SMS significantly widened the urban-rural gap in the informed choice of family planning. This finding was supported by other studies [36–38]. The possible justification behind this could be related to the fact that the availability and utilization of mobile phones were limited in rural areas [39–41]. Besides, rural resident women are less educated and could not read and understand messages sent to them via SMS [42, 43]. This study demonstrated that the frequency of listening to the radio was a significant variable that widen the urban-rural gap in the informed choice of family planning methods. The possible reason might be there is better access and coverage of mass media including Radio in urban areas because of improved energy access [44, 45]. Additionally, urban women are more literate to understand and internalize the information streamed through radio programs than their rural counterparts.

## Strengths and limitations

In this study, nationally representative data were used, allowing for better generalizability. The design effect due to the hierarchical nature of the samples was accounted for by using the STATA command "svyset" for each descriptive and analytical analysis. But, as this article is analyzed using secondary data, important behavioral and other socio-demographic variables might have declined model performance. The EDHS survey relies on respondents' self-reports, which may be subject to recall bias Furthermore, some women may be embarrassed to disclose all the details (self-report) concerning FP because it is a sensitive subject. Conclusion

In this study, visit to private health facilities, women's age group between 25 to 39 and visit to health facilities in the last 12 months are factors that significantly widen urban -rural gap of family planning. On the other hand, factors such as Access to FP related mobile SMS and frequency of listening to radio were found to narrow the urban -rural gap of informed choice of family planning. These findings suggest that to narrow residential inequalities of informed choice, tailoring interventional plans based on age, health seeking behavior and family planning promotion via media are effective strategies.

The spatial irregularity of not informed choice of family planning identified in this research needs further research to identify the factors behind these geographic inequalities using geographic weighted regression.

## Acknowledgments

The authors acknowledge MEASURE DHS for permitting us to access and download the Ethiopian 2016 DHS datasets.

## Author Contributions

**Conceptualization:** Abiyu Abadi Tareke, Mohammedjud Hassen Ahmed, Masresha Derese Tegegne.

**Data curation:** Abiyu Abadi Tareke, Bayley Adane Takele, Mohammedjud Hassen Ahmed.

**Formal analysis:** Abiyu Abadi Tareke, Bayley Adane Takele, Masresha Derese Tegegne.

**Funding acquisition:** Abiyu Abadi Tareke.

**Investigation:** Abiyu Abadi Tareke, Mohammedjud Hassen Ahmed.

**Methodology:** Abiyu Abadi Tareke, Mohammedjud Hassen Ahmed.

**Project administration:** Abiyu Abadi Tareke, Mohammedjud Hassen Ahmed.

**Resources:** Abiyu Abadi Tareke, Mohammedjud Hassen Ahmed, Masresha Derese Tegegne.

**Software:** Abiyu Abadi Tareke, Bayley Adane Takele.

**Supervision:** Abiyu Abadi Tareke, Habitu Birhan Eshetu.

**Validation:** Abiyu Abadi Tareke.

**Visualization:** Abiyu Abadi Tareke, Bayley Adane Takele, Habitu Birhan Eshetu.

**Writing – original draft:** Abiyu Abadi Tareke, Bayley Adane Takele, Habitu Birhan Eshetu.

**Writing – review & editing:** Abiyu Abadi Tareke, Mohammedjud Hassen Ahmed, Masresha Derese Tegegne, Habitu Birhan Eshetu.

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
