## [Decision Letter · Decision Letter 0]

13 Apr 2023

PONE-D-23-01109Urban-rural inequalities and spatial arrangement of informed choice of FP in Ethiopia: further analysis of 2016 Ethiopian demographic health survey.PLOS ONE

Dear Dr. Tareke,

Thank you for submitting your manuscript to PLOS ONE. After careful consideration, we feel that it has merit but does not fully meet PLOS ONE’s publication criteria as it currently stands. Therefore, we invite you to submit a revised version of the manuscript that addresses the points raised during the review process.

We look forward to receiving your revised manuscript.

Kind regards,

Wudneh Simegn Belay, MSc

Academic Editor

PLOS ONE

Journal Requirements:

4. We note that Figure 3 in your submission contain map images which may be copyrighted. All PLOS content is published under the Creative Commons Attribution License (CC BY 4.0), which means that the manuscript, images, and Supporting Information files will be freely available online, and any third party is permitted to access, download, copy, distribute, and use these materials in any way, even commercially, with proper attribution. For these reasons, we cannot publish previously copyrighted maps or satellite images created using proprietary data, such as Google software (Google Maps, Street View, and Earth). For more information, see our copyright guidelines: http://journals.plos.org/plosone/s/licenses-and-copyright.

(1) You may seek permission from the original copyright holder of Figure 3 to publish the content specifically under the CC BY 4.0 license.  

Reviewers' comments:

Reviewer's Responses to Questions

**Comments to the Author**

1. Is the manuscript technically sound, and do the data support the conclusions?

Reviewer #1: Yes

Reviewer #2: Partly

2. Has the statistical analysis been performed appropriately and rigorously? 

Reviewer #1: Yes

Reviewer #2: Yes

3. Have the authors made all data underlying the findings in their manuscript fully available?

Reviewer #1: Yes

Reviewer #2: Yes

4. Is the manuscript presented in an intelligible fashion and written in standard English?

Reviewer #1: Yes

Reviewer #2: Yes

5. Review Comments to the Author

Reviewer #1: Review Comments to the Author

Using 2016 EDHS data the authors assessed urban-rural inequalities and spatial arrangement of informed choice of family planning in Ethiopia. The paper reported that spatial configuration of not informed choice was clustered and magnitude of informed choice was 12 percent higher in urban residents compared to rural residents.

Title:

It is not recommended to use abbreviation on the title: so better you write full form.

Abstract: correct editorial error such as not capitalizing the first letter of new sentence and capitalizing the middle word ‘’Descriptive’’. The same editorial issues are there under discussion.

This sentence is incomplete and what authors want to say is not clear -More specifically, place of FP offers i.e., private health facilities (-16%), listening to the radio programs (+12%), age 35 to 49 years (-9.3%) and 25 to 34 years (+9.2%) (p<0.05.

Check grammar for the following sentence:

In this study, visit private health facilities, women’s age group between 25 to 39 and visit to health facilities in the last 12 months factors that significantly widen urban - rural gap of family planning.

You said: Besides, the spatial arrangement of not informed choice of family planning is not regular. To express spatial distribution the term random/non-random vs clustered are better rather than using not regular.

Introduction:

Check grammar issue in the following sentence:

•Informed choice of family planning means that when women choose a family planning methods, all available information regarding side effects, what to in case of side effects, and possible alternative methods

•Write full form followed by abbreviations in bracket first time abbreviations appear in manuscript

Check grammar issue in the following sentence:

•Moreover, the current study identified the spatial distribution and identified hotspots areas where women who had uninformed family planning choices.

Methods:

•Even if the authors used secondary data some sections of methods were absent. Please add subsections on study area, design, population and data source

•Editorial issue: don’t capitalize letters in the middle of the sentence “Visit”

Results:

•In figure 1 title mention time component, you can use Year EDHS was released. Similarly for table 1 title is incomplete address where and when component.

•Under the following section the authors wrongly cited Figure 1 replace it with Figure 2 because there are no p value in Figure 1 “The spatial arrangement of the uninformed choice of FP”

•Similarly in the paragraph under figure 2, omit figure 2 (wrongly cited)

Discussion

•Check grammar issue in the following sentence:

In this study, being in the age group 35 to 49 years was significantly narrowed the urban and rural gap in the informed choice of family planning where as being in the age group 25 to 34 found to widen the urban -rural gap of family planning compared to age group 15 to 24 year.

•Try to include strength and limitations of this study

Conclusions

•The first sentence is unnecessary

Declarations

You specified only single abbreviations but there are a lot of abbreviations inside document. So include all abbreviations-SNNP, NGO, FP……

References

Some references are too old (16=1996, 24=2007, 28=2001) . Try to use recent publications

Reviewer #2: Dear editors.

Thank you very much for this opportunity to review this paper. This paper assessed urban-rural disparities and spatial distribution of informed choice of FP in Ethiopia, which is crucial for plociy makers and program managers to identify intervention on inequality.

The following are my suggestions for the improvement of this manuscript:

General comments

The authors don't include page numbers and lines in the document, making it difficult to give suggestions and comments by page and line

Make sure to leave space before each bracket or parenthesis

There is typo and spelling error thoughout the document

Introduction.

The introduction part must be rewritten orked and organised in a logical way. The aouthors must highlights what lloks the condition in Africa, then in Ethiopia. Incorporate national family planning strategy and initiatives, as well as health service accessibility and distribution by residence. Describe the efforts and gaps in providing access to quality family planning service to everyone who needs it. It should be discussed that how much govt has spent on family planning and benefit in terms of preventing unintended pregnancy and highrsik fertility behaviors

FP: to be defined first

Method

EDHS is used two stage stratified study design so there is a need for adjustment of cluster and weight while doing multivariate analysis. Need to elaborate the design of the study and whether any type of stratification has been carried out? In addition, the authors did a decomposition analysis but did not elaborate on it in the method section. Which residence coded as “0” or “1”? it need more details

Independent variables

Be consistent, for some variable athors mentioned the coatergy but form sone not yet

Results

Present the frequency of Table 1 by residence, then the total and weighted frequencies are enough.

Decomposition analysis

The authors stated that "the composition factors of region, religion, and place of residence were not included in the multivariable composition analysis because they had a p-value greater than 0.2 at the bivariable decomposition analysis". How could residence be an explanatory variable here? It was an urban-rural disparity study, hence the grouping variable in this case is residence.

In table one, I didn't notice a variable “place for the FP offer” to be classified as government or private, but in the decomposed table, I did?????

Correct the FF typoerror

“in the f informed choice”

“urbanand rural”

“by 9.3%.. Oppositely”

Discusion

It may be helpful to have a short summary of important findings in the first paragraph. The authors did not discuss limitations of the study such as the self-reported nature of the outcome informed choice of FP.

It is surprising that the composition of private health facilities had a narrowing effect, which indicates the composition of private health facilities was higher in rural areas, which is far from reality. Most private health facilities in Ethiopia are concentrated in urban areas.

Conclusion

State the the policy implications of this study based on the finding.

6. PLOS authors have the option to publish the peer review history of their article (what does this mean?). If published, this will include your full peer review and any attached files.

Reviewer #1: No

Reviewer #2: No

---

## [Author Response · Author response to Decision Letter 0]

13 May 2023

Background: Ethiopia has made satisfactory progress in improving maternal and child health over the past two decades. The introduction of family planning through informed choice is one of the main strategies to improve maternal and child health. However, this positive progress may have masked the significant urban-rural disparities in informed choice for family planning.

Objective: To identify factor contributing to observed urban-rural disparities and to determine the spatial distribution of informed family planning choices in Ethiopia.

Methods: The study used information from 3,511 (1) women currently using contraceptives (rural-2685 and urban-826)c from the most recent Ethiopian demographic health survey cross-sectional data. Spatial and descriptive, bivariable, and multivariable logit-based decomposition analysis methods were used.

Results: The spatial configuration of uninformed choice was clustered. The primary cluster (LLR=34.8, p-value<0.001) was located at the southern portion of Amhara region that covers east & west Gojjam, south Gondar and south Wollo administrative zones. The magnitude of informed choice was 12 percent higher in urban residents compared to rural residents. Urban-rural gap was attributed to variations in characteristics (74%). Place of family planning offer i.e., private health facility, being aged between 35 and 49 years, and having visited to health facility in the last 1 year are found decrease the urban-rural gap of informed family planning choice by 15%, 9% and 5% respectively. Conversely, being aged between 25 and 34 years, being a listener to radio has increased the gap by 9% and 12% respectively. 

Conclusion: The variables being private health facility visitors, being aged between 25 and 39 years and having visited health facilities in the last one year are found to increase the gap of informed family planning choices between urban and rural residents Besides, the spatial distribution of uninformed family planning choices is non-random.

---

## [Decision Letter · Decision Letter 1]

5 Jul 2023

PONE-D-23-01109R1Urban-rural inequalities and spatial arrangement of informed choice of Family planning in Ethiopia: further analysis of 2016 Ethiopian demographic health survey.PLOS ONE

Dear Dr. Tareke,

Thank you for submitting your manuscript to PLOS ONE. After careful consideration, we feel that it has merit but does not fully meet PLOS ONE’s publication criteria as it currently stands. Therefore, we invite you to submit a revised version of the manuscript that addresses the points raised during the review process.

We look forward to receiving your revised manuscript.

Kind regards,

Wudneh Simegn, MSc

Academic Editor

PLOS ONE

Journal Requirements:

Additional Editor Comment:

Author’s contributions form must be corrected as per the journal guideline.

Reviewers' comments:

Reviewer's Responses to Questions

**Comments to the Author**

1. If the authors have adequately addressed your comments raised in a previous round of review and you feel that this manuscript is now acceptable for publication, you may indicate that here to bypass the “Comments to the Author” section, enter your conflict of interest statement in the “Confidential to Editor” section, and submit your "Accept" recommendation.

Reviewer #2: All comments have been addressed

2. Is the manuscript technically sound, and do the data support the conclusions?

Reviewer #2: Yes

3. Has the statistical analysis been performed appropriately and rigorously? 

Reviewer #2: Yes

4. Have the authors made all data underlying the findings in their manuscript fully available?

Reviewer #2: Yes

5. Is the manuscript presented in an intelligible fashion and written in standard English?

Reviewer #2: Yes

6. Review Comments to the Author

Reviewer #2: Dear Editor, thank you for allowing me to review “Urban-rural inequalities and spatial arrangement of informed choice of Family planning in Ethiopia: further analysis of 2016 Ethiopian demographic health survey”. After I reviewed the revised manuscript, I got that the author/authors had corrected all the raised issue. However, see carefully the age group in the abstract section particularly at result and conclusion section (35 to 49 VS 25 to 39).

7. PLOS authors have the option to publish the peer review history of their article (what does this mean?). If published, this will include your full peer review and any attached files.

Reviewer #2: No

---

## [Author Response · Author response to Decision Letter 1]

6 Jul 2023

Authors’ response to reviews

Title: Urban-rural inequalities and spatial arrangement of informed choice of family planning in Ethiopia: further analysis of 2016 Ethiopian demographic health survey.

Authors:

Abiyu Abadi Tareke (abiyu20010@gmail.com)

Bayley Adane Takele 

, Mohammedjud Hassen Ahmed

 Masresha Derese Tegegne

 Habitu Birhan Eshetu

Version: 2

 Date: May 13, 2023

Point by point response for editors/reviewers’ comments

Manuscript number: PONE-D-23-01109 

Dear editor/reviewer:

Dear all,

We express our profound appreciation for the insightful and productive feedback that you have provided. Your invaluable comments have significantly enriched the quality of the manuscript, and have greatly augmented our expertise in the realm of scientific paper writing. The authors have diligently taken into account each of the comments and queries raised by the editors and reviewers, and have responded to them in a targeted manner. Our comprehensive point-by-point rejoinders to all the comments and questions can be found in the subsequent pages. In addition, an accompanying supplementary document has been enclosed, which showcases the modifications made in detail, using the track changes feature. We also made some change to fix grammatical error in some paragraphs. 

Review Comments to the Author

Reviewer’s comment: see carefully the age group in the abstract section particularly at result and conclusion section (35 to 49 VS 25 to 39).

Authors’ response: dear reviewer we are here because of your insight and deep review to this manuscript. Sorry for this typographical error. After checking the result written in the regression table, we changed the phrase “being aged between 25 and 39 years” to “being aged between 35 and 49 years”.

---

## [Editor Report · Decision Letter 2]

12 Jul 2023

Urban-rural inequalities and spatial arrangement of informed choice of Family planning in Ethiopia: further analysis of 2016 Ethiopian demographic health survey.

PONE-D-23-01109R2

Dear Dr. Tareke,

We’re pleased to inform you that your manuscript has been judged scientifically suitable for publication and will be formally accepted for publication once it meets all outstanding technical requirements.

Kind regards,

Wudneh Simegn, MSc

Academic Editor

PLOS ONE
---

## [Editor Report · Acceptance letter]

11 Aug 2023

PONE-D-23-01109R2 

Urban-rural inequalities and spatial arrangement of informed choice of Family planning in Ethiopia: further analysis of 2016 Ethiopian demographic health survey. 

Dear Dr. Tareke:

I'm pleased to inform you that your manuscript has been deemed suitable for publication in PLOS ONE. Congratulations! Your manuscript is now with our production department. 

Kind regards, 

on behalf of

Dr. Wudneh Simegn 

Academic Editor

PLOS ONE